# Covalent organic framework nanofluidic membrane as a platform for highly sensitive bionic thermosensation

Pengcheng Zhang[1], Sifan Chen[1], Changjia Zhu[1,2], Linxiao Hou[1], Weipeng Xian[1], Xiuhui Zuo[1], Qinghua Zhang[1], Lin Zhang [1], Shengqian Ma [2✉] & Qi Sun [1✉]

Thermal sensation, which is the conversion of a temperature stimulus into a biological response, is the basis of the fundamental physiological processes that occur ubiquitously in all organisms from bacteria to mammals. Significant efforts have been devoted to fabricating artificial membranes that can mimic the delicate functions of nature; however, the design of a bionic thermometer remains in its infancy. Herein, we report a nanofluidic membrane based on an ionic covalent organic framework (COF) that is capable of intelligently monitoring temperature variations and expressing it in the form of continuous potential differences. The high density of the charged sites present in the sub-nanochannels renders superior perms-electivity to the resulting nanofluidic system, leading to a high thermosensation sensitivity of 1.27 mV K$^{-1}$, thereby outperforming any known natural system. The potential applicability of the developed system is illustrated by its excellent tolerance toward a broad range of salt concentrations, wide working temperatures, synchronous response to temperature stimulation, and long-term ultrastability. Therefore, our study pioneers a way to explore COFs for mimicking the sophisticated signaling system observed in the nature.

[1] Zhejiang Provincial Key Laboratory of Advanced Chemical Engineering Manufacture Technology, College of Chemical and Biological Engineering, Zhejiang University, Hangzhou, China. [2] Department of Chemistry, University of North Texas, Denton, TX, USA. ✉email: shengqian.ma@unt.edu; sunqichs@zju.edu.cn

The ability of living organisms to perceive environmental temperature is crucial for maintaining normal life processes[1–3]. In mammals, thermal stimuli are converted into electrochemical potentials via thermosensitive transient receptor potential (thermo-TRP) ion channels, which are then translated into an action potential, such as a sensation of pain, by the thermoreceptor nerve cells[4–6]. Significant efforts have been devoted for implanting thermally active molecules in artificial systems to regulate their thermosensation activity[7–13]. These systems exhibit reversible thermal responsiveness; however, they only work when the temperature reaches the thermal transition temperature, thereby limiting their application to a narrow temperature window.

Recent developments in the principles and applications of nanofluidic transport allow us to better understand and mimic the function of biological pores[14–16]. Surface charge induced electrostatic ion screening is one of the essential features of nanofluidic systems. The electrostatic forces from the fixed charges on the nanochannels counteract the tendency of the co-ions to get transported along the direction of driving forces, such as temperature and concentration gradients, resulting in an unequal distribution of charged species between the nanochannels and solution phase[17,18]. A potential difference could therefore be established in the steady state at the interphase, which can be recorded under open-circuit conditions ($V_{oc}$, Eq. 1), where $c^\alpha$, $T^\alpha$ and $c^\beta$, $T^\beta$ are the concentrations and temperatures of the two solutions, and $t_+$, $R$, and $F$ are the transference number of the cation, gas constant, and Faraday constant, respectively.

$$V_{oc} = -2t_+\left(\frac{RT^\beta}{F}\ln c^\beta - \frac{RT^\alpha}{F}\ln c^\alpha\right) \qquad (1)$$

According to this equation, the thermoelectric response to temperature changes can be principally expressed as a continuum of changes in a potential, which is similar to what is observed in nature. It can also be inferred that the cation permselectivity of the nanofluidic device is a pivotal parameter for the generation of a significant potential difference in response to temperature changes under otherwise identical conditions. One of the critical factors that determine the permselectivity of a charged nanofluidic system is the channel dimension; it acts as an effective ion screen only when the inside radius of the channel is relatively smaller than the thickness of the electrical double layer of the solution. Therefore, to improve the thermosensation performance, it is of great importance to develop nanofluidic systems that allow the simultaneous manipulation of the pore size and fixed charged sites[19–22].

Recent advancements in materials science have rendered opportunities to tackle these issues, and a prime example is the advent of two-dimensional (2D) COFs. Their unparalleled versatility allows many aspects to be custom-designed to control the pore size and pore environment, thereby lending credence to their prospects for various task-specific applications[23–37]. Moreover, the 2D sheets in 2D COFs can naturally produce a membrane via stacking, which is driven by π-stacking in the third dimension, thereby providing a direct transport channel through the pores. Therefore, COFs can serve as an ideal platform for the construction of a membrane-scale nanofluidic device (Fig. 1)[38–46].

A high ionic density is beneficial for generating a unipolar ionic environment, and the Debye screening length is a characteristic length that depends on the ionic strength of the solution (higher the ionic strength, smaller the Debye length). Hence, it could be concluded that a highly charged COF material with a small pore channel would be preferable for achieving a high permselectivity over a wide range of ion concentrations[47]. Therefore, to imitate the thermo-TRP ion channels observed in nature, a COF

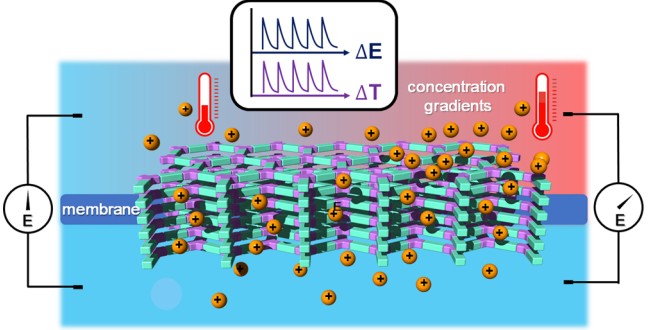

**Fig. 1 Ionic 2D COF nanofluidic membrane for thermosensation.** Schematic illustration of the principle of ionic covalent organic framework nanofluidic membrane as a bionic thermometer.

synthesized by the condensation of 1,3,5-triformylphloroglucinol (Tp) and triaminoguanidine hydrochloride (Tag) was chosen for the construction of nanofluidic systems (Fig. 2a, b). This is because such a COF has a high density of intrinsic charge and the smallest pore size among the 2D COFs reported to date[42,48,49]. With these attributes, the resulting membrane exhibits synchronous transmembrane diffusion potential difference in response to the temperature gradient across the membrane over a broad range of ion concentrations and temperature windows with a sensitivity up to 1.27 mV K$^{-1}$, thereby outperforming the natural thermosensation systems. With further research and development, we expect the modified system to be useful for artificial intelligence applications.

## Results

**Membrane fabrication and characterization.** The COF-based membrane was grown via acid-catalyzed interfacial polymerization. Acetic acid and Tag dissolved in an aqueous phase were physically separated from the Tp dispersed in a mixture of ethyl acetate and mesitylene to allow the formation of COF active layers exclusively on the polyacrylonitrile (PAN) support (TpTag-COF/PAN), which was placed at the liquid–liquid interface using a homemade diffusion cell (Fig. 2c, and Supplementary Fig. 1 and 2). The use of PAN as a support is mainly based on the considerations that it is flexible, which can increase the operability of the resulting membrane, and it is hydrophilic and negatively charged, which can lower the transmembrane energy of cations. Over the course of 3 days, a yellow film was formed on the side of the PAN support facing the organic phase. Scanning electron microscopy images revealed smooth, crack-free, continuous film surfaces that contoured the underlying PAN support with a height profile of ~100 nm (Supplementary Fig. 3 and 4). The successful formation of the β-ketoenamine structures of the resulting COF membranes was confirmed by the attenuated total reflection infrared (ATR-IR) analysis, which displayed a new peak for C=C at 1601 cm$^{-1}$. The stretching signals of the primary amine ($\nu_{N–H}$ ~3320 cm$^{-1}$ and 3220 cm$^{-1}$) and aldehyde ($\nu_{c=O}$ = 1645 cm$^{-1}$) were not detected, indicating that no unreacted monomers were trapped (Supplementary Fig. 5)[42,48–50]. The zeta potentials of the TpTag-COF/PAN membrane and PAN were −40.88 mV and −51.1 mV, respectively, in the presence of 1 mM KCl (pH ~7). The powder X-ray diffraction analysis of the free-standing TpTag-COF membrane revealed crystalline structures with two relatively broad peaks at the 2θ values of ~11° and ~28°, which are assignable to the (100) and (001) facets, respectively (Supplementary Fig. 6). To elucidate the constitution of the framework, a theoretical simulation was performed using Materials Studio, which displayed a pore size of 0.8 nm along the c-axis (Fig. 2b).

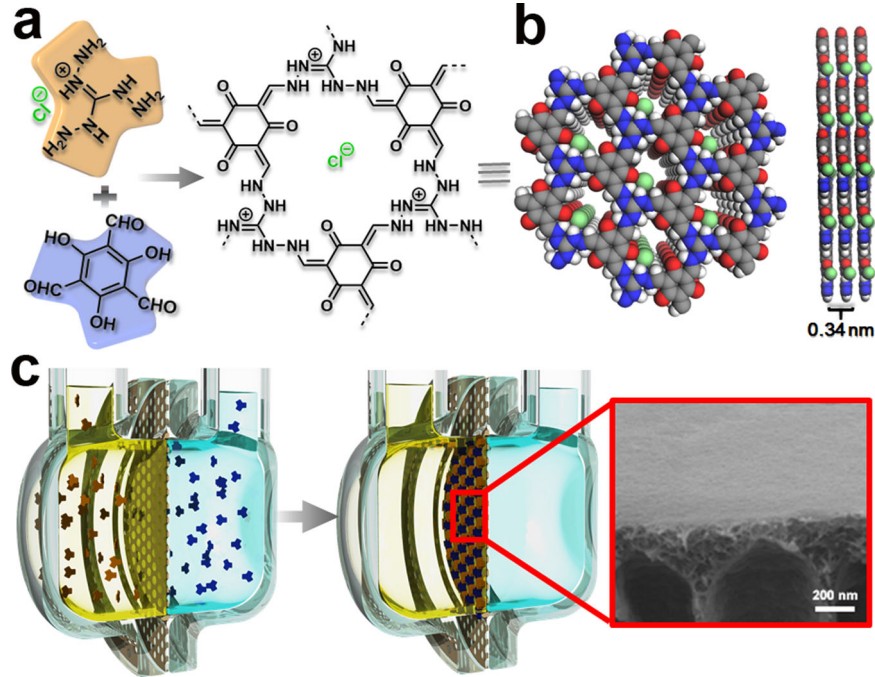

**Fig. 2 Membrane synthesis. a** Synthetic scheme of TpTag-COF through the condensation of Tp and Tag. **b** Graphic view of the eclipsed AA stacking structure of TpTag-COF (blue, N; gray, C; red, O; white, H; green, Cl). **c** Schematic illustration of TpTag-COF/PAN and the corresponding SEM image.

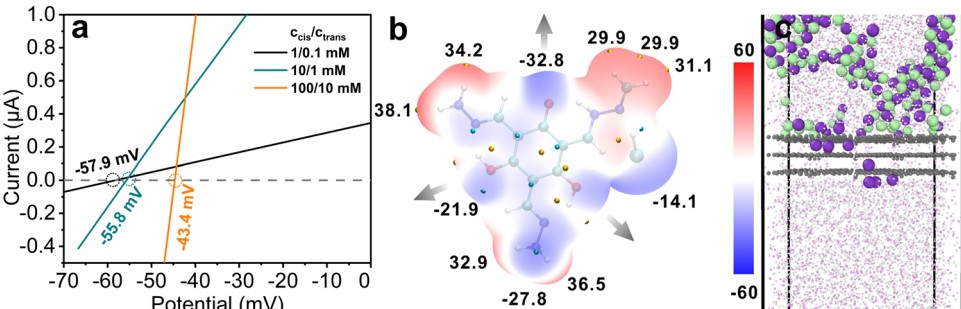

**Fig. 3 Ion permselectivity evaluation. a** I−V plots under various asymmetric KCl solutions separated by TpTag-COF/PAN (black, 1/0.1 mM; olive, 10/1 mM; orange, 100/10 mM). **b** Electrostatic potential (ESP) mapped van der Waals surface of the fragment of TpTag-COF and area percent in each ESP range. Significant surface local minima and maxima of ESP are represented as red and blue spheres and labeled by black texts. The direction of pore content is highlighted marked by grey arrows. **c** The schematic diagram of ion transport behavior through TpTag-COF subnanochannels at 500 ps. The MD simulation revealed that TpTag-COF/PAN shows a higher K⁺ ion transport activity than Cl⁻ ion (purple, K; green, Cl; red, O; white, H; gray, TgTag-COF layers).

**Ion permselectivity evaluation**. To determine the permselectivity of TpTag-COF/PAN, the reversal potentials under various KCl concentration ranges were evaluated. The choice of KCl is because of the very similar bulk mobilities of $K^+$ and $Cl^-$ ions[51]. The X-intercepts ($V_r$) of the plots of the current versus voltage (I–V) gave reversal potentials of −57.9, −55.8, and −43.4 mV for cis/trans = 1 mM/0.1 mM, 10 mM/1 mM, and 100 mM/10 mM KCl aqueous solutions, respectively (Fig. 3a, cis refers to the side facing to the COF active layer). The negative values were the first indication of the preferential passage of $K^+$ over the $Cl^-$ ions through TpTag-COF/PAN. The $K^+/Cl^-$ permeability ratios derived from the Goldman–Hodgkin–Katz equation (Eq. 2) are 244, 76, and 12 for the aforementioned solution systems, respectively, which validates the high cation permselectivity of TpTag-COF/PAN. In this equation, $a$, $R$, $F$, and $T$ refer to the ion activity, gas constant, Faraday constant, and absolute temperature, respectively. The preferred transport of cations over anions across TpTag-COF/PAN was also observed in other electrolytes, including NaCl, LiCl, and $MgSO_4$. Under a concentration gradient of 100 mM/10 mM NaCl, LiCl, and $MgSO_4$, the $Na^+/Cl^-$,

$Li^+/Cl^-$, and $Mg^{2+}/SO_4^{2-}$ permeability ratios were calculated to be 10, 8, and 3.5, respectively. The discrepancy in permselectivity can be rationalized by the different diffusion rates of the ions (Supplementary Fig. 7)[52].

$$\frac{P_{K^+}}{P_{Cl^-}} = \frac{a_{Cl^-,cis} \cdot \exp\left(-V_r F/RT\right) - a_{Cl^-,trans}}{a_{K^+,cis} - a_{K^+,trans} \cdot \exp\left(-V_r F/RT\right)} \quad (2)$$

These results contradict the traditional model of an ionic channel with fixed cationic sites, across which the transportation of anions is usually preferred. We ascribed the overscreened surface charge to the strong hydrogen bonding interactions between the framework and $Cl^-$ ions, which affects the overall electronic configuration. This leads to a negative pore surface, and hence promotes the $K^+$ ion transport[53,54]. To rationalize this assumption, we performed quantum density functional theory computations to map the charge distribution. The minimum energy geometries of TpTag-COF demonstrated that there was a large negative partial charge on the O species (Fig. 3b and

Supplementary Fig. 8). To further understand the ion transport behavior, molecular dynamics (MD) simulations were performed (see details in the Supplementary Information). The $4 \times 4 \times 3$ TpTag-COF layers were sandwiched by a KCl aqueous solution and deionized water. In addition to containing 2000 water molecules, there were 200 KCl molecules in the KCl aqueous solution. Starting from a random configuration, followed by a 1 fs production run, the system was stopped after a run of 1 ns. The schematic diagram of the system at 500 ps, as shown in Fig. 3c, indicates that only the water molecules and $K^+$ ions can translocate from the KCl aqueous solution to the COF layers, thereby confirming the higher transport activity of cations than that of anions through TpTag-COF (Supplementary Fig. 9).

**Thermoelectric response evaluation.** Based on these results, we further investigated the thermoelectric response of TpTag-COF/PAN (see the schematic illustration of the experimental setup in Supplementary Fig. 10). To perform this study, the membrane was placed between the two temperature-controlled chambers of a permeation cell. The TpTag-COF active layer always faced the chamber with a lower temperature. A temperature gradient was induced by the brief heating of one chamber using a heat rod. The temperature gradient was controlled to within ~10 K. This is because the variation of the activity coefficient and electrochemical potential of the ions with changes in temperature can be ignored in this range. The thermal stimulus drives the ion transport, which is screened by the membrane, resulting in a potential difference at the boundary of the membrane that can be recorded under open-circuit conditions. The real-time temperature difference between the two chambers was monitored by thermocouples, and the open-circuit voltage ($V_{oc}$) was detected using Ag/AgCl electrodes. To establish the connection between the change in $V_{oc}$ ($\Delta V_{oc}$) relative to the initial value and the change in solution temperature ($\Delta T$), the evolution of $\Delta V_{oc}$ and $\Delta T$ with time were recorded using an electrochemical workstation and a temperature recorder, respectively (see details of theoretical derivation of thermoelectric response across membranes in Supplementary Information, Supplementary Equations 1–21). The ion permselectivity of a nanofluidic system is dependent on the solution concentration; therefore, we first evaluated the impact of KCl concentrations on the performance of the thermoelectric response of TpTag-COF/PAN. To specifically investigate the sensitivity of thermosensation, the two chambers were filled with symmetric KCl aqueous solutions in the range of 0.5–100 mM, and similar trajectories were observed for the time evolution of $\Delta V_{oc}$ and $\Delta T$ (Fig. 4a). To calculate the sensitivities of thermosensation, we plotted $\Delta V_{oc}$ against $\Delta T$ according to Eq. 3, where $t_+$, $R$, $F$, $\Delta T$, and $a^T$ are the transference number of the cation, gas constant, Faraday constant, change in temperature, and ion

activity at temperature T, respectively.

$$\Delta V_{oc}(T, a) = -2t_+ \frac{R}{F} \Delta T ln a^T \qquad (3)$$

The resulting curves were well fitted with the linear model, with correlation coefficients higher than 0.99 (Fig. 4b). The sensitivities of thermosensation, which were derived from the slopes, were found to be in the range of 1.27–0.40 mV $K^{-1}$ over the KCl concentrations of 0.5–100 mM; the former decreased as the latter increased (Fig. 4c). For further verification, TpTag-COF/PAN was benchmarked against the other reported systems. At the KCl concentrations under 10 mM, SIM/PET hybrid and PET conical nanochannels exhibited thermosensation sensitivities of 0.71 mV $K^{-1}$ and 0.44 mV $K^{-1}$, respectively, which were inferior to that of TpTag-COF/PAN (0.79 mV $K^{-1}$)[18]. Under otherwise identical conditions, the PAN support afforded this value of 0.51 mV $K^{-1}$, indicative of the role of the TpTag-COF active layer (Supplementary Fig. 11). To highlight the role of continuous regular pore channels of the COF, we compared the thermosensation sensitivity of the BtTag/PAN membrane, synthesized by condensation of 1,3,5-benzenetricarbonyl trichloride (Bt) and Tag on PAN, with TpTag-COF/PAN. BtTag/PAN afforded a thermosensation sensitivity of 0.63 mV $K^{-1}$ using 10 mM of KCl as an electrolyte, which is inferior to that of TpTag-COF/PAN (Supplementary Fig. 12). Furthermore, the ion permeability of TpTag-COF/PAN was over two orders of magnitude higher than that in BtTag/PAN. Collectively, we attribute the superior ion permselectivity of TpTag-COF/PAN to its small, ordered channels and high surface charge density. To gain an in-depth insight of the membrane properties, the connection between the transmembrane diffusion potential ($\phi_{diff}$) and temperature change was established, where $\Delta\phi_{diff}$ can be obtained by subtracting $\Delta E_{redox}$ ($E_{redox}$: redox potential) from the detected $\Delta V_{oc}$. A linear fit of the calculated $\Delta\phi_{diff}$ vs $\Delta T$ was obtained according to Equation (Supplementary Equation 16); $t_+$ was calculated to be 0.967 (Supplementary Fig. 15), which is greater than that of the SIM/PET hybrid nanochannels (0.90), further validating the superior transmembrane permselectivity of TpTag-COF/PAN. Remarkably, the $t_+$ values of TpTag-COF/PAN were maintained over a KCl concentration range of 0.5–50 mM ($t_+ = 0.956$–$0.995$); on increasing the KCl concentration to 100 mM, only a slight decrease was observed in the $t_+$ value ($t_+ = 0.908$) (see details in Supplementary Table 1). These results can be rationalized by comparing the Debye screen lengths of these solutions and the channel size of TpTag-COF (Supplementary Table 2); the Debye screening length ($\lambda_D$) of 100 mM KCl is 0.96 nm, which is close to the channel size of TpTag-COF, resulting in a less effective ion screening efficiency.

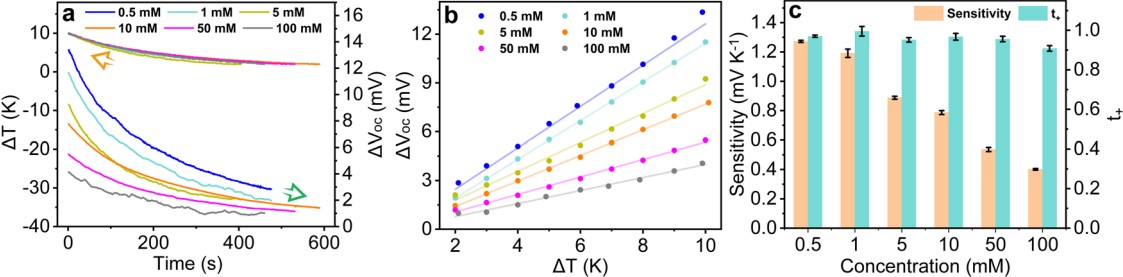

**Fig. 4 Thermoelectric response evaluation. a** Thermoelectric responses of TpTag-COF/PAN placed between various symmetric KCl aqueous solutions (blue, 0.5 mM; cyan, C; dark yellow, 5 mM; orange, 10 mM; magenta, 50 mM; grey 100 mM). The synchronous time evolution $\Delta V_{oc}$ in response to the solution temperature changes ($\Delta T$, the overlapped curves above the orange arrow) was recorded. **b** The linear fits of $\Delta V_{oc}$ against $\Delta T$ according to Eq. 2; all the fits have $R^2$ values higher than 0.99 (blue, 0.5 mM; cyan, C; dark yellow, 5 mM; orange, 10 mM; magenta, 50 mM; grey, 100 mM). **c** The corresponding thermosensation sensitivity and $t_+$ values were derived from Eq. 2 (average of three different batch experiments, orange, selectivity; cyan, $t_+$).

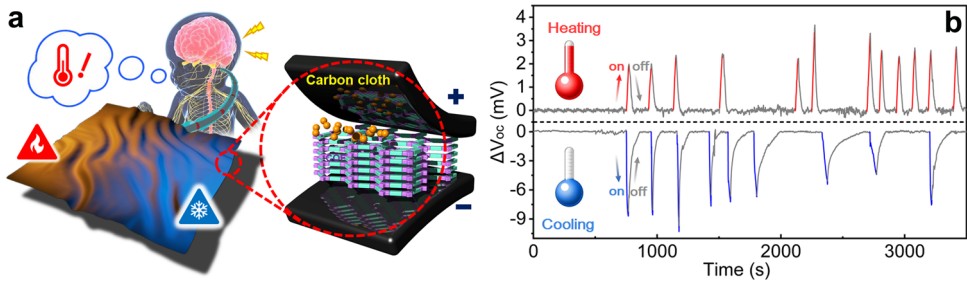

**Fig. 5 Thermosensation monitoring. a** Schematic illustration of how the smart textile was designed with the ability of thermosensation. **b** Real-time measured potential changes, corresponding to the imposed temperature gradients (red, heating on; grey, heating/cooling off; blue, cooling on).

1 mM KCl solution was chosen for further study, considering that it has a high sensitivity and is less affected by the impurities from the environment. Given the importance of the stability of a thermosensation system for practical applications, continuous heating and cooling cycles were introduced to investigate the repeatability, which revealed that the variation of $V_{oc}$ with $\Delta T$ was maintained for at least 20 cycles (Supplementary Fig. 16), with an average value of 1.15 mV K$^{-1}$. As an additional metric, we analyzed the response speed. As shown in Supplementary Fig. 17, a rapid change in temperature led to an instantaneous change in $V_{oc}$, attaining a response speed of 109% (average of 20 cycles shown in Supplementary Fig. 16), as calculated according to Eq. 4:

$$S = \frac{t_T}{t_{V_{oc}}} \times 100\% \tag{4}$$

In this equation, $t_T$ is the time taken to reach a specific temperature detected by two immersed microthermometers, and $t_{Voc}$ is the time taken for $V_{oc}$ to reach the maximum value during heating. A value higher than 100% is partly because of the delay of the thermocouple, which, indicates the instantaneous response of TpTag-COF-PAN to the temperature change. This was further validated by the profiles of the continuous and synchronous time evolution curves of $\Delta V_{oc}$ and $\Delta T$, with their discrepancy in time more considerable along with the cycling.

To further validate the applicability of the developed thermosensation system, its thermoelectric responses toward other temperature windows were evaluated. The system appeared to exhibit very high sensitivity in response to the varying temperature windows. The damping of ion permselectivity was not observed, thereby affording sensitivities of 1.19 and 1.15 mV K$^{-1}$ in the temperature ranges of 15–25 °C and 35–45 °C, respectively. This indicates the applicability of the system over a wide temperature window (Supplementary Fig. 18 and 19).

Considering these results, the system was further investigated to mimic the thermosensation behavior of aquatic organisms, wherein salt concentration gradients usually exist between their body fluids and environment. For this purpose, the thermosensation performance of TpTag-COF/PAN was evaluated in the presence of asymmetric KCl aqueous solutions. The concentration gradient and hot stimulation drive the ion transport in opposite directions; therefore, we placed the COF active layer facing the concentrated solution, and the temperature gradient was imposed on the low-salt concentration side. The initial $V_{oc}$ values were measured to be 116 mV for the system with a concentration gradient of 200 mM/10 mM (representative of the salt gradient between organisms in freshwater and their living environment)[55], corresponding to the $t_+$ value of 0.943, indicating high cation selectivity. After introducing a temperature gradient of approximately 10 K, a linear dependence of $\Delta V_{oc}$ and $\Delta T$ was also observed, with thermosensation sensitivity of 0.78 mV K$^{-1}$ (Supplementary Fig. 20).

**Fabrication of wearable temperature sensor**. Based on these results, we explored the potential of TpTag-COF/PAN for the design of wearable devices with thermosensation ability. To act as a sensor, the membrane was placed between the two pieces of a carbon cloth soaked in a symmetric KCl solution (1 mM). A temperature gradient was induced by briefly heating or cooling one piece of the carbon cloth, and was measured using the method described above. The resulting change in potential was detected by Ag/AgCl electrodes. After the system returned to thermal equilibrium, another temperature gradient was imposed to detect the dynamic potential variations. Figure 5 displays the real-time output potential response of the thermosensation system, proving its sensitivity and repeatability. Moreover, there is a clear difference in the magnitude of the change in output potential in response to the different temperature gradients introduced. Collectively, the developed thermosensation system displays great potential for use in the design of wearable temperature sensors and beyond.

## Discussion

In summary, we have described a conceptual application of ionic COFs as highly effective thermal sensors. This study, which capitalizes on the established ion permselectivity of a nanofluidic system, demonstrates that the ionic COF-based sub-nanochannels can mimic the thermo-TRP ion channels observed in nature. Considering our preliminary experimental results, which suggest that a combination of TpTag-COF/PAN and carbon cloth can exhibit changes in potential in response to varying temperatures, we are currently developing an artificial skin with the ability to detect the temperature of the environment. This work represents a highly encouraging step in the development of thermo-responsive devices.

## Methods

Commercially available reagents were purchased in high purity and used without further purification. Triaminoguanidine hydrochloride (Tag) and 1,3,5-tri-formylphloroglucinol (Tp) were purchased from Jilin Chinese Academy of Sciences-Yanshen Technology Co., Ltd. The polyacrylonitrile (PAN) ultrafiltration membrane was obtained from Sepro Membranes Inc. (Carlsbad, CA, USA) with a molecular weight cutoff of 40000 Da. Scanning electron microscopy (SEM) was performed on a Hitachi SU 8000. X-ray powder diffraction (XRD) patterns were measured with a Rigaku Ultimate VI X-ray diffractometer (40 kV, 40 mA) using CuKα ($\lambda$ = 1.5406 Å) radiation. FT-IR spectra were recorded on a Nicolet Impact 410 FTIR spectrometer. The Zeta potentials of the membranes were performed on SurPASS 3 (1 mM KCl, pH = 7).

## Data availability

The authors declare that all the data supporting the findings of this study are available within the article (and Supplementary Information files), or available from the corresponding author on reasonable request. Source data are provided with this paper.

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

## Acknowledgements

The authors acknowledge the National Science Foundation of China (21776241, 2196116074, and 222071132) and the Fundamental Research Funds for the Central Universities (2019XZZX003-04) for the financial support of this work. Partial support from the Robert A. Welch Foundation (B-0027) is also acknowledged (SM). We thank Dr. Xianfeng Yi from the Wuhan Institute of Physics and Mathematics for his help with the electrostatic potential calculation. We are also grateful to Prof. Bin Su from the Department of Chemistry, Zhejiang University, for his valuable suggestions.

## Author contributions

Q.S. and S.M conceived and designed the research. P.Z. performed the synthesis. P.Z., S.C., C.Z., L.H., W.X., and X.Z. carried out the tests. Q.Z. and L.Z. provided valuable suggestions. All authors participated in drafting the paper, and gave approval to the final version of the manuscript.

## Competing interests

The authors declare no competing interests.
