## [Peer Review File · Nature Communications]

REVIEWER COMMENTS:

Reviewer #1 (Remarks to the Author):

This work reports the fabrication of thermal sensors based on ionic covalent organic frameworks (COFs). Although COFs have not been reported previously to use as enablers in thermal sensors, the idea to use charged nanochannels to detect the temperature change is not new. Moreover, the COF-based thermal sensors developed in this work do not show much better performances than others (e.g. Ref. 18), and I do not see any advantages of their thermal sensors over previously reported ones. I am sorry I am not in the position to support the acceptance of this work in high-tier journals like NC although this is a solid work containing both experimental investigations and simulations.

Reviewer #2 (Remarks to the Author):

This study reported a highly sensitive COF based thermosensation, which is very interesting. The reported results are very good. Considering overall performance of the thermosensation, its publication in Nature Communications is suggested.

1 What about the value of cation permselectivity compared to the reported results?

2 I guess it's not appropriate to extend the obtained results without further experiments. For example, it's indeed that K⁺ can translocate from the solution to the COF layer. However, we are not sure this is also applied to other cations. So, it's not safe to conclude that the results also confirm the higher activity of cations than that of anions unless some other cations were also tested.

3 The caption of Fig. 4 was departed from the figures by some text.

4 The membrane can act as a sensor, however, the best application might not in smart textile fabrication because it is not common to have textile applications in the environment of KCl solution.

5 It seems that the sensitivities of thermosensation are just slightly higher than the reported results (0.79 mV K⁻¹ compared to 0.71 mV K⁻¹), which is not so exceptional.

6 In process of COF/PAN fabrication, which solution was first put in the diffusion cell? Is there a possibility that one solution will cross the PAN before the other solution was filled?

7 It would be better to use the concentration gradients similar to the aquatic organisms when the author mimics the thermosensation behavior of aquatic organisms

8 The contribution of the arrayed pores of COF to the performance such as sensitivities of the thermosensation was not clearly explained. Some control experiments might could elucidate this.

Reviewer #3 (Remarks to the Author):

This manuscript reported a nanofluidic membrane based on an ionic covalent organic framework (COF) that can intelligently monitor temperature variations. It is quite a novel and interesting work to apply COF-based membrane to bionic thermal sensation. The study is well designed and the conclusion can be well supported by the experimental data presented. So, this manuscript is recommended for publication in Nature Communications after the following revisions are

performed.

1. Why did the authors choose PAN to combine with COF material?
2. The authors said that “it acts as an effective ion screen only when the inside radius of the channel is relatively smaller than the thickness of the electrical double layer of the solution.” So how do we compare the pore size of TpTag-COF and the thickness of the electrical double layer of the solution.
3. It is suggested that the authors provide the detailed derivation process of Equation 1.
4. The number of equation in line 3 of page 3 is missing.
5. The expression in line 19 of page 3 is confusing. The dispersion forces are not exactly the same as π -stacking.

Dr. Shengqian Ma
Professor
Robert A. Welch Chair in Chemistry

Department of Chemistry
University of North Texas
1508 W Mulberry St
Denton, TX 76201
E-mail: Shengqian.Ma@unt.edu
Phone: 940-369-7137
Fax: 940-565-4318
Website: <http://www.chemistry.unt.edu/~sqma/>

Date: January 6, 2021

We appreciate the constructive comments and suggestions from the reviewers, and we have revised the manuscript accordingly as detailed in the responses below. The corresponding changes have been highlighted in yellow in the main text and supplementary information.

Reviewer #1:

Comment 1: This work reports the fabrication of thermal sensors based on ionic covalent organic frameworks (COFs). Although COFs have not been reported previously to use as enablers in thermal sensors, the idea to use charged nanochannels to detect the temperature change is not new. Moreover, the COF-based thermal sensors developed in this work do not show much better performances than others (e.g. Ref. 18), and I do not see any advantages of their thermal sensors over previously reported ones. I am sorry I am not in the position to support the acceptance of this work in high-tier journals like NC although this is a solid work containing both experimental investigations and simulations.

Response: We appreciate the reviewer for taking the time to review our work. First of all, we appreciate the reviewer's positive comment, "this is a solid work containing both experimental investigations and simulations." Seemingly, this work only advanced COFs as a new platform to mimic the function of thermo-TRP in nature, which has been demonstrated using other materials (Ref. 18). Our work has made great strides from the earlier research, where we have designed a membrane material with intrinsic charged sites and sub-nano channel size. In the previous work, Su et al. developed a mesoporous silica-based nanofluidic system (the ionic sites were introduced by adjusting the solution pH), wherein the thermoelectric response to temperature changes was expressed as changes in potential, which is similar to what is observed in nature. From this seminal work, it was concluded that the cation permselectivity of the nanofluidic device is a pivotal parameter for the generation of significant potential in response to the temperature changes under otherwise identical conditions. One of the critical factors that determine the permselectivity of a charged nanofluidic system is the channel dimension, and only when the channel size is smaller than the thickness of the electrical double layer (EDL) of the solution, it can screen ions effectively. The pore size of the developed SIM/PET is around 2.3 nm, and its cation ion transference number (t_+) is calculated to be 0.9 using 10 mM KCl as an electrolyte ($\lambda_D \approx 3$ nm). It can be inferred that by further increasing the electrolyte concentration, the thermosensation sensitivity would be greatly dropped as the thickness of EDL over the channel size. Therefore, to extend the application of the thermosensation system, it is very important to develop nanofluidic systems that allow the

simultaneous manipulation of the pore size and fixed charged sites. This work demonstrates how 2D COFs with tailorable pore structures display the right combination of properties to serve as a new platform for highly sensitive bionic thermosensation. The high density of the charged sites present in the sub-nanochannels renders superior permselectivity to the resulting nanofluidic system. Under otherwise identical conditions used by Su et al., the t_+ value was calculated to be 0.967 for TpTag-COF/PAN, which greatly improved permselectivity. To achieve better performance, we optimized the test conditions. A t_+ value of 0.995 and a thermosensation sensitivity up to 1.27 mV K^{-1} were obtained. Furthermore, we also extend the application of the thermosensation system as demonstrated by the design of smart textiles with thermosensation ability. In addition, the electrolytes used in this study can be acid, alkaline, or neutral, not like SIM/PET, where the basic solution is required for achieving a negatively charged pore channel, which inevitably limits its application. In summary, we greatly improve upon the initial studies from Su et al. by providing additional to design a better material with a refined application and consider its potential for use in smart textile applications for wearables and beyond. Therefore, we believe our work can meet the urgent requirements for Nature Communications, as also concurred by both Reviewer 2 and Reviewer 3, and a wide readership will benefit from our discovery.

Reviewer #2:

Comment 1: This study reported a highly sensitive COF based thermosensation, which is very interesting. The reported results are very good. Considering overall performance of the thermosensation, its publication in Nature Communications is suggested.

Response: We appreciate the reviewer's high comments and support of our work.

Comment 2: What about the value of cation permselectivity compared to the reported results?

Response: We thank the reviewer's comment. Under otherwise identical conditions, the t_+ value was calculated to be 0.967, which is greater than that of the SIM/PET hybrid nanochannels (0.90), validating the superior transmembrane permselectivity of TpTag-COF/PAN. Remarkably, due to the sub-nano channel and high charge density, the t_+ values of TpTag-COF/PAN were maintained over KCl concentrations ranging from 0.5 mM to 50 mM (0.956-0.995); on increasing the KCl concentration to 100 mM, only a slight decrease in the t_+ value (0.908) was observed.

Comment 3: I guess it's not appropriate to extend the obtained results without further experiments. For example, it's indeed that K^+ can translocate from the solution to the COF layer. However, we are not sure this is also applied to other cations. So, It's not safe to conclude that the results also confirm the higher activity of cations than that of anions unless some other cations were also tested.

Response: We thank the reviewer for the constructive comment. The preferred transport of cations over anions across TpTag-COF/PAN was also observed in other electrolytes, including NaCl, LiCl, and MgSO_4 . Under a concentration gradient of 100 mM/10 mM NaCl, LiCl, and MgSO_4 , the Na^+/Cl^- , Li^+/Cl^- , and $\text{Mg}^{2+}/\text{SO}_4^{2-}$ permeability ratios were calculated to be 10, 8, and 3.5,

respectively. The discrepancy in permselectivity can be rationalized by the different diffusion rates of the ions

Comment 4: The caption of Fig. 4 was departed from the figures by some text.

Response: We thank the reviewer for pointing this out. We have adjusted the position of Fig.4.

Comment 5: The membrane can act as a sensor, however, the best application might not in smart textile fabrication because it is not common to have textile applications in the environment of KCl solution.

Response: We thank the reviewer for valuable criticism. The showcase study required further development to act as a wearable temperature sensor. To avoid the leakage of KCl solution, the textile can be wrapped with a dense and flexible film or be assembled into a cell. To make the demonstrated potential applications sounds reasonable, we have revised the smart textile into a wearable temperature sensor.

Comment 6: It seems that the sensitivities of thermosensation are just slightly higher than the reported results (0.79 mV K^{-1} compared to 0.71 mV K^{-1}), which is not so exceptional.

Response: We thank the reviewer for the comment. According to the seminal work from Su et al., it was concluded that the cation permselectivity of the nanofluidic device is a pivotal parameter for the generation of significant potential in response to the temperature changes under otherwise identical conditions. One of the critical factors that determine the permselectivity of a charged nanofluidic system is the channel dimension, and only when the channel size is smaller than the thickness of the electrical double layer (EDL) of the solution, it can screen ions effectively. The pore size of the developed SIM/PET is around 2.3 nm, and its cation ion transference number (t_+) is calculated to be 0.9 using 0.1 M KCl as an electrolyte ($\lambda_D \approx 3 \text{ nm}$). It can be inferred that by further increasing the electrolyte concentration, the thermosensation sensitivity of SIM/PET would be greatly dropped as the thickness of EDL over its channel size. Therefore, to extend the application of the thermosensation system, it is very important to decrease the channel size. In this work, a COF-based membrane with a channel size of 0.8 nm has been fabricated. The high density of the charged sites present in the sub-nanochannels renders superior permselectivity to the resulting nanofluidic system. Under otherwise identical conditions used by Su et al., the t_+ value was calculated to be 0.967 for TpTag-COF/PAN, which greatly improved permselectivity. To achieve better performance, we optimized the test conditions. A t_+ value of 0.995 and a thermosensation sensitivity up to 1.27 mV K^{-1} were obtained.

Comment 7: In process of COF/PAN fabrication, which solution was first put in the diffusion cell? Is there a possibility that one solution will cross the PAN before the other solution was filled?

Response: We thank the reviewer for the comment. We usually fill the two chambers simultaneously using two pipettes. However, due to the large capillary force of the nanopore of the ultrafiltration membrane (30-50 nm), the solution can still be held for a long time if only one side of the diffusion cell was filled.

Comment 8: It would be better to use the concentration gradients similar to the aquatic organisms when the author mimics the thermosensation behavior of aquatic organisms

Response: We thank the reviewer for the valuable suggestion. Per the reviewer's suggestion, we have evaluated the thermosensation performance of TpTag-COF/PAN under the concentration gradients of the body fluid of organisms in freshwater (0.2 M) and their living environment (0.01 M), affording a thermosensation sensitivity of 0.78 mV K^{-1} .

Comment 9: The contribution of the arrayed pores of COF to the performance such as sensitivities of the thermosensation was not clearly explained. Some control experiments might elucidate this.

Response: We thank the reviewer for the insightful comment. To highlight the role of continuous regular pore channels of the COF, we compared the thermosensation sensitivity of the BtTag/PAN membrane, synthesized by condensation of 1,3,5-benzenetricarbonyl trichloride (Bt) and Tag on PAN, with TpTag-COF/PAN. BtTag/PAN afforded a thermosensation sensitivity of 0.63 mV K^{-1} using 10 mM of KCl as an electrolyte, which is inferior to that of TpTag-COF/PAN. Furthermore, the ion permeability of TpTag-COF/PAN was over two orders of magnitude higher than that in BtTag/PAN. Collectively, we attribute the superior ion permselectivity of TpTag-COF/PAN to its small, ordered channels and high surface charge density.

Reviewer #3:

Comment 1: This manuscript reported a nanofluidic membrane based on an ionic covalent organic framework (COF) that can intelligently monitor temperature variations. It is quite a novel and interesting work to apply COF-based membrane to bionic thermal sensation. The study is well designed and the conclusion can be well supported by the experimental data presented. So, this manuscript is recommended for publication in Nature Communications after the following revisions are performed.

Response: We appreciate the reviewer's high comments and support of our work.

Comment 2: Why did the authors choose PAN to combine with COF material?

Response: We thank the reviewer for the comment. The use of PAN as the support is mainly based on the considerations that it is flexible, which can increase the operability of the resulting membrane, and it is hydrophilic and negatively charged, which can lower the transmembrane energy of cations.

Comment 3: The authors said that "it acts as an effective ion screen only when the inside radius of the channel is relatively smaller than the thickness of the electrical double layer of the solution." So how do we compare the pore size of TpTag-COF and the thickness of the electrical double layer of the solution.

Response: We thank the review for the comment. The thickness of electrical double layer is determined by the type and the concentration of electrolyte, which can be calculated using the following equation

$$\lambda_D = \left(\frac{\varepsilon_r \varepsilon_0 k_B T}{2 N_A e^2 I'} \right)^{1/2}$$

where ε_r and ε_0 are the vacuum and relative permittivity, respectively, k_B is the Boltzmann constant, T is the absolute temperature, e is the elementary charge, N_A is the Avogadro number, and I is the ionic strength of the solution.

Comment 4: It is suggested that the authors provide the detailed derivation process of Equation 1.

Response: We thank the reviewer for the comment. The detailed derivation process of Equation 1 has been provided in the Supplementary Information in the section of theoretical derivation of thermoelectric response.

Comment 5: The number of equation in line 3 of page 3 is missing.

Response: We thank the reviewer for pointing this out. We have included the equation number.

Comment 6: The expression in line 19 of page 3 is confusing. The dispersion forces are not exactly the same as π -stacking.

Response: We thank the reviewer for pointing this out. To avoid confusion, we have reword this sentence.

Again, we thank the reviewer for the constructive comments and suggestions, which have made our manuscript much improved.

Sincerely,

Shengqian Ma, PhD
Professor and Welch Chair in Chemistry

REVIEWERS' COMMENTS

Reviewer #2 (Remarks to the Author):

The authors have considered all the concerns from me. The quality of the manuscript has been highly improved after the corrections and additions done by the authors. I have no further reservations to the acceptance of the manuscript.

Reviewer #3 (Remarks to the Author):

The author has revised the manuscript well according to the questions and suggestions raised by reviewers. I believe this revised manuscript is now suitable for publication in Nature Communications.

Dr. Shengqian Ma
Professor
Robert A. Welch Chair in Chemistry

Date: June 16, 2019

Department of Chemistry
University of North Texas
1508 W Mulberry St
Denton, TX 76201
E-mail: Shengqian.Ma@unt.edu
Phone: 940-369-7137
Fax: 940-565-4318
Website: <http://www.chemistry.unt.edu/~sqma/>

We greatly appreciate the positive comments and constructive suggestions from the reviewers. We have revised the manuscript accordingly as detailed in the responses below, and the corresponding changes have been highlighted in yellow.

Reviewer #1

Comment: The authors have considered all the concerns from me. The quality of the manuscript has been highly improved after the corrections and additions done by the authors. I have no further reservations to the acceptance of the manuscript.

We are grateful to the reviewer for taking the time to evaluate our work and support from the reviewer. The concern raised by the reviewer has been addressed.

Reviewer #2

Comment 1: The author has revised the manuscript well according to the questions and suggestions raised by reviewers. I believe this revised manuscript is now suitable for publication in Nature Communications.

We thank the reviewer for taking the time to evaluate our work and support from the reviewer.

Again we thank the reviewers for the constructive comments and suggestions, which have made our manuscript much improved.

Sincerely,

Shengqian Ma, PhD
Professor and Welch Chair in Chemistry